# Clinical Presentation of Hepatocellular Carcinoma in African Americans vs. Caucasians: A Retrospective Analysis

**Hrishikesh Samant [1], Kapil Kohli [1], Krunal Patel [2], Runhua Shi [3], Paul Jordan [1], James Morris [1], Annie Schwartz [3] and Jonathan Steven Alexander [1,3,*]**

1   Section of Gastroenterology and Hepatology, Departments of Medicine, Ochsner-LSU Health Sciences Center in Shreveport, Shreveport, LA 71103, USA; hrishikesh.samant@ochsner.org (H.S.); kkohli@lsuhsc.edu (K.K.); pjorda1@lsuhsc.edu (P.J.); jmorri2@lsuhsc.edu (J.M.)
2   Lane Gastroenterology, Zachary, LA 70791, USA; krunalhp@gmail.com
3   Department of Molecular and Cellular Physiology, LSU Health Sciences Center Shreveport, Shreveport, LA 71130, USA; rshi@lsuhsc.edu (R.S.); ams004@lsuhs.edu (A.S.)
*   Correspondence: jalexa@lsuhsc.edu; Tel.: +1-318-675-4151

**Abstract:** Hepatocellular carcinoma (HCC) remains an important form of cancer-related morbidity and mortality in the U.S. and worldwide. Previous U.S.-based studies on survival suggest ethnic disparities in HCC patients, but the complex interplay of multiple factors that contribute are still incompletely understood. Here we considered the influences of risk factors contributing towards HCC survival, including ethnic background, over ten years at a premier academic medical center with a majority (57.20%) African American (AA) population. Retrospective HCC data were collected from 2008–2018 at LSUHSC-Shreveport, an urban tertiary medical center. Data included demographics, comorbidities, liver disease characteristics, and tumor parameters. Statistical analysis was performed using Chi Square and one-way ANOVA. Results: 229 HCC patients were identified (male 78.6%). The mean HCC age at diagnosis was 61 years (SD = 7.3). Compared to non-Hispanic Caucasians (42.7%), AA patients (57.2% of total) were older at presentation, had more frequent diabetes/dyslipidemia/NAFLD (45 (34.3%) compared with 19 (19.3%) in non-Hispanic Caucasians, *p* = 0.02), and had a larger HCC burden at diagnosis. We conclude that compared to white patients, despite having similar BMI and MELD scores and rates of portal vein thrombosis, AA patients with HCC in our cohort were older at presentation, had a significantly increased incidence of modifiable metabolic risk factors including diabetes, higher AFP values, increased incidence of gallstones, and larger sized HCCs, and were more likely to be outside Milan criteria. These findings have important prognostic and diagnostic implications for developing a more targeted HCC surveillance program.

**Keywords:** hepatocellular carcinoma; ethnicity; race; overall survival; survival disparity

## 1. Introduction

Hepatocellular carcinoma (HCC) causes ~800,000 deaths annually [1] and is the fourth leading cause of cancer-related death [1]. Hepatocellular cancer (HCC) is the fifth most frequently diagnosed cancer in men, and the ninth most commonly diagnosed cancer in women [2]. The National Cancer Institute Surveillance (SEER) Database stated that HCC incidence rates increased 3.1 percent per year from 2008–2012 [3].

HCC exhibits unique geographic, sex, and age distributions that may reflect etiologic factors. In total, 82% of HCC cases occur in developing countries that have high rates of chronic HBV infection, (Southeast Asian/African countries vs. US). Four major etiologic factors associated with HCC have been established: chronic viral infection (HCV, HBV), alcoholism, non-alcoholic steatohepatitis (NASH), and food contaminants (aflatoxin).

Prior studies analyzing liver cancer burden have focused on single countries or regions, single years, or a subset of the most common etiologies, such as HBV and HCV. In this study we report results of epidemiology and demography on primary liver cancer incidence,

mortality, and disease distribution for AA vs. non-Hispanic Caucasian populations from 2008 to 2018 by sex, and on the burdens of liver cancer attributable to HCV, alcohol, and a remaining "other" group that encompasses residual causes. We considered African Americans in comparison with Caucasians in our clinical practice. Nationally, African Americans make up ~14% of the US population but are often under-represented in studies on race-based presentation of disease and survival. However, because our region has an approximate population makeup of ~50% AA and 50% white, we are better able to consider such contributions. Here we have evaluated such contributions to HCC. This study was conducted to evaluate the factors responsible for these differences which can help us to serve our community better.

## 2. Materials and Methods

### 2.1. Patient Selection

This retrospective study used data collected from HCC patients treated at Louisiana State University Health Sciences Center, Shreveport between 2008 and 2018. Data were collected only for non-Hispanic Caucasians and AA because Hispanic Caucasians, Asians, and Native Americans (American Indians/Alaska natives) represented less than 4% of our patient population, so were not evaluated because of low-sample sizes. Survival time was extracted from date of last known follow-up or last imaging when unknown. The follow-up cut-off date was set as 31 October 2018. In our study we considered various etiological, demographic, and treatment-specific parameters, as mentioned in Tables 1 and 2. These parameters were decided based on the literature for HCC diagnosis and treatment.

**Table 1.** Patient characteristics for the whole study cohort at HCC presentation.

| Variable | Variable | Caucasian N (%) | African American N (%) | *p*-Value |
|---|---|---|---|---|
| Age (years) | 60 | 56 (57.14%) | 48 (36.64%) | 0.003 |
| | Age (years) > 60 | 42 (42.86%) | 83 (63.36%) | |
| Comorbidities: | Comorbidities: group 1 | 52 (53.06%) | 49 (37.40%) | 0.02 |
| | Comorbidities: group 2 | 19 (19.30%) | 45 (34.35%) | |
| | Comorbidities: group 3 | 27 (27.55%) | 37 (28.24%) | |
| AFP | AFP value 0–20 (group 1) | 42 (45.65%) | 30 (23.81%) | 0.001 |
| | AFP value 21–500 (group 2) | 31 (33.70%) | 49 (38.89%) | |
| | AFP value > 500 (group 3) | 19 (20.65%) | 47 (37.30%) | |
| HCC | HCC size < 3 cm | 49 (50%) | 51 (38.93%) | 0.02 |
| | HCC size 3–5 cm | 26 (26.53%) | 27 (20.61%) | |
| | HCC size > 5 cm | 23 (23.47%) | 53 (40.46%) | |
| Milan Score | Milan score (meeting criteria) | 60 (66.67%) | 60 (49.18%) | 0.01 |
| | Milan score (outside the criteria) | 30 (33.33%) | 62 (50.82%) | |

**Table 1.** *Cont*.

| Variable | Variable | Caucasian N (%) | African American N (%) | *p*-Value |
|---|---|---|---|---|
| Gallstone | Gallstone present | 19 (19.38%) | 46 (35.11%) | 0.03 |
| | Gallstone absent | 69 (70.40%) | 75 (57.25%) | |
| Presentation grouping | Biliary sludge | 6 (6.12%) | 6 (4.58%) | 0.02 |
| | Cholecystectomy | 18 (19.14%) | 13 (10.23% | |
| | Gallbladder present | 76 (80.85%) | 114 (89.76%) | |
| Sex | Male | 72 (73.47%) | 108 (82.44%) | 0.1 |
| | Female | 26 (26.53%) | 23 (17.56%) | |
| Marital Status | Married | 26 (26.53%) | 34 (25.95%) | 0.1 |
| | Unmarried | 47 (47.96%) | 77 (58.78%) | |
| | Divorced | 25 (25.51%) | 20 (15.27%) | |
| Serum Creatinine | Serum Creatinine < 1.3 (mg/dL) | 89 (90.82%) | 108 (83.08%) | 0.1 |
| | Serum Creatinine > 1.3 (mg/dL) | 9 (9.18%) | 22 (16.92%) | |
| Total bilirubin | Total bilirubin < 1 (mg/dL) | 38 (38.78%) | 64 (49.23%) | 0.1 |
| | Total bilirubin > 1 (mg/dL) | 60 (61.22%) | 66 (50.77%) | |
| Hepatic encephalopathy | Hepatic encephalopathy controlled on medications (group 1) | 23 (23.47%) | 25 (19.08%) | 0.09 |
| | Hepatic encephalopathy uncontrolled on medications (group 2) | 16 (16.33%) | 11 (8.40%) | |
| | Hepatic encephalopathy absent (group 3) | 59 (60.20%) | 95 (75.52%) | |
| Statin | Statin being used | 14 (14.29%) | 30 (22.90%) | 0.1 |
| | Statin not being used | 84 (85.71%) | 101 (77.10%) | |
| Aspirin | Aspirin being used | 22 (22.45%) | 41 (31.30%) | 0.1 ns |
| | Aspirin not used | 76 (77.55%) | 90 (68.70%) | |
| Diagnosed by | Diagnosed by ultrasound | 24 (25.53%) | 45 (35.71%) | 0.1 |
| | Diagnosed by CT or MRI | 70 (74.47%) | 81 (64.29%) | |

**Table 1.** *Cont*.

| Variable | Variable | Caucasian N (%) | African American N (%) | *p*-Value |
|---|---|---|---|---|
| ultrasound | Lesion on screening ultrasound | 19 (45.24%) | 44 (58.67%) | 0.1 |
| | Lesion not seen on ultrasound | 23 (54.76%) | 31 (41.33%) | |
| BCLC stage | BCLC stage A | 10 (11.24%) | 11 (9.02%) | 0.1 |
| | BCLC stage B | 33 (37.08%) | 29 (23.77%) | |
| | BCLC stage C | 36 (40.45%) | 62 (50.82%) | |
| | BCLC stage D | 10 (11.24%) | 20 (16.39%) | |

**Table 2.** Summary of findings on univariate analysis between Caucasian and AA patients with HCC at presentation.

| Variable | Mean | SD |
|---|---|---|
| Age (years) | 61.07 | 7.32 |
| Platelet count (k/mm3) | 148.37 | 97.87 |
| Serum Sodium (Meq/L) | 137 | 3.83 |
| Total bilirubin (mg/dL) | 1.77 | 2.23 |
| Creatinine (mg/dL) | 1.08 | 1.06 |
| INR | 1.26 | 0.31 |
| Albumin (mg/dL) | 3.13 | 0.65 |
| MELD score | 12.34 | 5.19 |
| HCC size (largest in cm) | 5.04 | 4.10 |
| ALT (IU) | 91.13 | 97.57 |

The groupings of comorbidities are designated as follows: group 1 patients had no comorbidities; group 2 patients had dyslipidemia, diabetes, or non-alcoholic fatty liver disease (NAFLD); group 3 patients had hypertension, autoimmune disease, or malignancy.

AFP values were grouped as follows: group 1 had AFP values between 0 and 20, group 2 had AFP values between 21 and 500, and group 3 had AFP values greater than 500. Bialecki [4] states that: "AFP >400–500 ng/mL is considered diagnostic for HCC, although fewer than half of patients may generate levels that high." Zhang et al. [5], in a meta-analysis, showed that AFP values in excess of 200 ng/mL are associated with HCC. Therefore, we also selected values of AFP >200 as an upper cut-off value.

Hepatic encephalopathy (HE) was grouped as follows: group 1 had hepatic encephalopathy controlled by medication; group 2 had uncontrolled HE, even with medication; group 3 had no HE.

The Barcelona clinic liver cancer (BCLC) staging is defined as follows: Stage A includes single or up to 3 nodules ≤3 cm along with preserved liver function with ECOG PS 0. Stage B is defined as multinodular disease with preserved liver function with ECOG PS 0. Stage C shows portal invasion with extrahepatic spread and preserved liver function with ECOG PS 1 to 2. Stage D is end stage liver function (Child Turcotte Pugh Stage C) with ECOG PS 3 to 4.

*2.2. Statistical Analysis*

Descriptive statistics are reported as means with standardized deviations (SD) for continuous variables, and percentages for categorical variables. The Student's t-test was used to compare continuous variables between groups. The Pearson Chi-square test was used for comparing groups for categorical variables. Multivariate Cox regression was used to assess the effect of factors on survival of these patients. Statistical analyses were performed by using SAS 9.4 for Windows (SAS Institute Inc., Gary, NC, USA). A two-tailed $p < 0.05$ was considered statistically significant.

**3. Results**

*3.1. Patient Population*

In our study, a total of 229 patients diagnosed with HCC from 2008 to 2018 were included. Among these, 42.80% were non-Hispanic Caucasians and 57.20% were AA (summarized in Tables 1 and 2). Non-Hispanic Caucasians were older in general: 57.4% were ≤ 60 years old, compared to 36.6% of AA patients ($p = 0.003$).

*3.2. Presence of Comorbidities*

In total, 53.06% of the non-Hispanic Caucasian patients were free of comorbidities (group 1); 19.30% were in group 2 and 27.55% were in group 3. By comparison, AA patients had fewer (37.40%) patients without comorbidities (group 1); 34.35% were in group 2 and 28.24% were in group 3. A significantly higher numbers of AA patients had comorbidities compared to non-Hispanic Caucasians ($p = 0.02$).

*3.3. AFP Values*

With regard to non-Hispanic Caucasian patients, 45.6% had AFP values of 0–20 (group 1); 33.8% had AFP values between 21 and 500 (group 2); and 20.6% had AFP values >500 (group 3). For AA patients, 30 (23.8%) had low AFP values (0–20), 38.9% had AFP values of 21–500 (group 2), and 37.30% had AFP values >500 (group 3). African Americans had significantly higher AFP values ($p = 0.001$) than whites.

*3.4. Tumor Size at Presentation*

In total, 50% of non-Hispanic Caucasians had an HCC sized <3 cm at diagnosis, and 38.9% of the AA population had the same. With respect to HCC size, 26.5% of whites had tumors sized 3–5 cm, whereas 20.61% of AA patients had the same. With regard to HCC tumors sized >5 cm, whites were found to have tumors of this size 23.4% of the time, whereas 40.4% of AA patients had tumors of this size. Non-Hispanic Caucasians were therefore more likely to have smaller tumors (<3 and 3–5 cm), and AA patients were more likely to have larger (>5 cm) tumors.

*3.5. Milan Score*

More non-Hispanic Caucasians were found to meet Milan criteria at diagnosis compared to AA patients ($p = 0.01$): 66.7% of non-Hispanic Caucasians met the Milan criteria, and 33.33% did not. In comparison, 49.1% AA patients met the Milan criteria, and 50.82% did not.

*3.6. Presence of Gallstones*

AA patients were significantly more likely (with $p = 0.03$) to have gallstones and less likely to have sludge compared to non-Hispanic Caucasians. With regard to gallstones, 19.38% of non-Hispanic Caucasians presented with gallstones, 70.40% did not have gallstones, and 6.12% had sludge. In comparison, 35.11% of AA patients had gallstones, 57.25% had no gallstones, and 4.58% had sludge.

### 3.7. Cholecystectomy

Non-Hispanic Caucasians were significantly (with $p = 0.02$) more likely to have had a cholecystectomy compared to AA patients. With regard to operations, 19.14% of non-Hispanic Caucasians had undergone a cholecystectomy, whereas 80.85% had not. In comparison, 10.23% of AA patients had undergone a cholecystectomy, and 89.76% had not.

### 3.8. Sex

In our study, approximately 78.6% of HCC patients were male and 21.4% were female overall. Of the patients with HCC, 73.47% and 82.44% were males who were non-Hispanic Caucasians and AA, respectively. However, this higher prevalence for males in AA HCC was found not to be statistically significant ($p = 0.1$ from non-Hispanic Caucasians); further studies are still needed to determine how sex and race influence trends in HCC screening.

### 3.9. Marital Status

Marital status is an important but indirect SES marker that may reflect a patient's ability to provide self-care. We found that non-Hispanic Caucasians and AA had similar marriage rates. AA were slightly more likely to be unmarried and non-Hispanic Caucasians were slightly more likely to be divorced ($p = 0.1$). Of non-Hispanic Caucasians, 26.53% were married at presentation, 47.96% were unmarried, and 25.5% were divorced. In AA, 25.9% were married, 58.78% were unmarried, and 15.2% were divorced.

### 3.10. Serum Creatinine

Non-Hispanic Caucasians were found to be slightly more likely to present with normal serum creatinine at presentation compared to AA patients: 90.8% of non-Hispanic Caucasian patients had creatinine <1.3 at presentation, compared to 83.08% AA patients ($p = 0.1$).

### 3.11. Bilirubin

Non-Hispanic Caucasians were also somewhat more likely to present with lower bilirubin: 38.78% of non-Hispanic Caucasians presented with total bilirubin <1 compared to 49.23% of AA patients ($p = 0.1$).

### 3.12. Hepatic Encephalopathy

Most (72.5%) AA patients did not have HE; more non-Hispanic Caucasian patients (60.2%) had HE, though the difference was not quite significant ($p = 0.09$). Of non-Hispanic Caucasian patients, 23.47% were in group 1 (HE controlled by medication), 16.33% were group 2 (HE not controlled by medication), and 60.20% had no HE. By comparison, in AA, 19.08% of patients had HE controlled by medication (group 1), 8.40% of patients had HE not controlled by medication (group 2), and 75.52% of patients had no HE (group 3).

### 3.13. Statin Use

AA patients had higher statin usage in our study. Of non-Hispanic Caucasian patients, 14.29% used statins, compared to 22.90% of AA patients ($p = 0.1$).

### 3.14. Aspirin Use

AA patients had higher aspirin use in our study: 22.45% of non-Hispanic Caucasian patients used aspirin, compared to 31.30% of AA patients ($p = 0.1$).

### 3.15. US/CT/MRI Diagnostics

Non-Hispanic Caucasians were more likely to be diagnosed with CT/MRI. AA were more likely to be diagnosed with screening ultrasound at presentation. In fact, 25.53% of non-Hispanic Caucasians were diagnosed with ultrasound and 74.47% with CT or MRI. Of AA patients, 35.71% of patients were diagnosed with ultrasound and 64.29% were diagnosed by CT or MRI ($p = 0.1$).

### 3.16. Discovery of Lesions by US

AA patients were more likely to have a positive screening ultrasound which led to the need for a second imaging test: 49% of the total cohort did not have guideline-based ultrasound screening, despite it being recommended by major medical societies.

In total, 45.24% of non-Hispanic Caucasian patients had lesions seen via screening ultrasound. In AA, 58.67% of patients had a lesion seen by US ($p = 0.1$).

BCLC stage at diagnosis: Caucasians were more likely to present at BCLC stages 1 and 2. AA patients were more likely to present at stages 3 and 4. In addition, 11.2% of non-Hispanic Caucasians were in BCLC stage A at presentation, 37% were in stage B, 40.4% were in stage C, and 11.2% were in stage D. For AA patients, 9% were in stage 1, 23.7% were in stage 2, 50.8% were in stage 3, and 16.3% were in stage 4 ($p = 0.1$).

### 3.17. Univariate Analysis

The following variables were found upon univariate analysis to be similarly distributed in both Caucasians and AA patients ($p > 0.1$): smoking, family history, ECOG performance status, BMI, etiology of cirrhosis, platelet count, serum sodium, INR, serum albumin, MELD score, Child Turcotte Pugh score, esophageal varices, ascites, number of lesions at diagnosis, portal vein invasion, metastasis workup, serum ALT, vitamin E use, site of metastasis, and medical insurance (Medicare, Medicaid, commercial, no insurance).

### 3.18. Multivariate Analysis

As detailed in Table 3, the patients who had lower ECOG PS, HCV infection, no alcohol use, no ascites, no portal vein thrombosis, no metastasis, an HCC size of 3–5 cm, and BCLC stage A, and who met the Milan criteria and had normal ALT did better than the patients with the opposite characteristics (with $p$ values and hazard ratios listed in Table 3). In multivariate analysis, we compared early disease compared to patients with more advanced disease. These findings highlight the need to improve the current surveillance guidelines, as delayed diagnosis contributes significantly to increased mortality. Higher ECOG scores (PS 2/3/4) significantly increased the risk of death as compared to ECOG of PS 0. Individuals with HCV using alcohol were 2.28 times more likely to die as compared to those with HCV without alcohol use.

**Table 3.** Summary of findings from multivariate analysis for patients with HCC at presentation.

| Variable | Sub-Category | Hazard Ratio | 95% Confidence Interval | *p*-Value |
|---|---|---|---|---|
| ECOG | PS 0 | 1 | | |
| | PS 1 | 2.29 | 0.83–6.28 | 0.1 |
| | PS 2/3/4 | 7.94 | 2.64–23.83 | 0.002 |
| HCV infection | without alcohol use | 1 | | |
| | with alcohol use | 2.28 | 1.02–5.12 | 0.04 |
| Ascites | Absent | 1 | | |
| | Controlled w/medications | 2.79 | 1.26–6.17 | 0.01 |
| | Uncontrolled w/medications | 3.50 | 1.47–8.34 | 0.004 |
| Malignant portal vein thrombosis | Absent | 1 | | |
| | Present | 5.63 | 1.41–22.47 | 0.01 |
| Number of locally directed therapies | 0 | 1 | | |
| | 1–2 | 0.41 | 0.22–0.78 | 0.006 |
| | 3–6 | 0.11 | 0.039–0.31 | 0.00004 |
| Metastasis | Absent | 1 | | |
| | Present | 2.37 | 1.15–4.90 | 0.01 |

**Table 3.** *Cont*.

| Variable | Sub-Category | Hazard Ratio | 95% Confidence Interval | *p*-Value |
|---|---|---|---|---|
| HCC | Size 3–5 cm | 1 | | |
| | Size > 5 cm | 2.32 | 0.76–7.05 | 0.1 |
| BCLC | Stage D | 1 | | |
| | Stage A | 0.16 | 0.01–1.52 | 0.1 |
| Milan criteria | Outside | 1 | | |
| | Within | 0.3 | 0.09–0.97 | 0.04 |
| ALT | Normal level | 1 | | |
| | High level | 1.6 | 0.8–3.07 | 0.1 |

## 4. Discussion

Since HCC-related mortality continues to increase in the U.S, ethnic disparities in overall survival have attracted significant attention [6]. In order to design improved and better-targeted screening programs aimed at reducing this disparity and its attendant mortality, concerted efforts have been made to identify the causes behind this pattern and its etio-pathologic causes [7,8].

Population-based studies in the US have identified racial and ethnic variations in the incidence of HCC, and they concluded that Asians/Pacific Islanders (APIs) have higher rates of HCC compared with other groups [9–11]. In a study using data from the Surveillance, Epidemiology, and End Results (SEER) program of the National Cancer Institute, the incidence of HCC was highest among Asians, nearly double that of Hispanic Caucasians (11 vs. 6.8 per 100,000 per year) and four times higher than that of non-Caucasians (2.6 per 100,000 per year) [10]. In another database analysis from the US, the incidence rates among APIs, African Americans, Native Americans/Alaska Natives, and non-Hispanic Caucasians were 7.8, 4.2, 3.2, and 2.6 per 100,000 persons, respectively [11]. Although HCC rates in the other groups are expected to remain the highest, HCC rates among non-Hispanic Caucasians are expected to increase.

In this study, we were able to demonstrate multiple significant differences in the presentation between AA and non-Hispanic Caucasian patients. These differences included AA patients at presentation (1) being older, (2) having increased incidences of modifiable metabolic risk factors, e.g., diabetes/dyslipidemia and NAFLD, (3) having a higher incidence of gallstones, (4) having larger HCCs and higher AFP values.

Worldwide, men are known to be more likely than women to develop HCC [12]. The disparity is more pronounced in high-incidence regions, where men are affected 2.1 to 5.7 times more frequently than women (mean 3.7:1). In Northern America, the incidence rates for males and females were 6.8 and 2.2 per 100,000 persons, respectively, in 2008 [12]. Although not fully understood, the differences in sex distribution are thought to be due to variations in hepatitis carrier status, exposure to environmental toxins, the trophic effect of androgens, and/or potentially protective effects of estrogen mediated through inhibition of interleukin 6 [13]. HCC seemed to have been more likely in AA males in our study, which is in agreement with previous studies, as listed above.

There were 56 (57.14%) non-Hispanic Caucasian patients <60 years at presentation vs. 48 (36.6%) AA patients, which is statistically significant *p* = 0.003. With age, the risk for HCC increases due to long standing liver disease. Several large prospective studies conducted in both Asia and Western Europe cited a mean age at presentation between 50 and 60 years [14–16]. The mean age of our cohort was 61.07 years (SD = 7.32), which is above that reported in previous studies. On the other hand, in our study, we noted a trend towards AA patients presenting at older ages compared to the Caucasian population, suggesting genetic factors may also play a role.

Overall, there was a statistically significant disparity in comorbidities among AA patients in comparison to the non-Hispanic Caucasians (*p* = 0.02): 19.39% of non-Hispanic Caucasian patients and 34.36% of AA patients had a history of Dyslipidemia, DM, or NAFLD. Epidemiologic studies suggest a possible link between diabetes mellitus and HCC [17–24], and multiple systemic reviews and meta-analyses have also found an association [25–27]. A systematic review that included 49 case-control and cohort studies estimated that the risk was increased by approximately 2.2-fold (risk ratio 2.2; 95% CI 1.7–3.0), although few studies adjusted for diet and obesity [25]. A meta-analysis of 14 prospective epidemiological studies also found increased risk of HCC among patients with diabetes (relative risk 1.9; 95% CI 1.2–2.3) [26]. In addition, a SEER database study found that the presence of the metabolic syndrome (defined by the presence of three of the following: elevated waist circumference/central obesity, dyslipidemia, hypertension, and impaired fasting glucose) was a risk factor for HCC (adjusted odds ratio 2.1) [28].

A large population-based cohort study confirmed the findings of the systematic review and meta-analysis. The study included 19,349 patients with newly diagnosed diabetes and 77,396 patients without diabetes [20]. The incidence of HCC was significantly higher among patients with diabetes compared with those without diabetes (21.0 vs. 10.4 per 10,000 person-years), with an adjusted hazard ratio (HR) of 1.7 (95% CI 1.5–2.0). The use of a thiazolidinedione or metformin was associated with a decreased risk of HCC among patients with diabetes (adjusted HR 0.56 and 0.49, respectively). However, associations between diabetes and HCC should be interpreted judiciously. In many cases, the onset of glucose intolerance results from the development of cirrhosis, so "diabetes" in this context may be a surrogate for cirrhosis, which increases the risk of HCC. In addition, many patients with diabetes also have nonalcoholic fatty liver disease (NAFLD), which has also been associated with increased risk of HCC.

There is growing evidence that NAFLD represents increasingly frequent underlying liver disease in patients with HCC [29–33]. It is likely that NAFLD contributes to HCC via cirrhosis, although the exact mechanisms(s) have yet to be determined, and at least one study found that HCC could occur in patients with NAFLD who did not have cirrhosis [33]. Another study found that HCC in NASH was associated with obesity, diabetes, hypertension, and male sex [32]. In view of the above findings, the statistically significant associations of diabetes, dyslipidemia, and NALFD with the AA cohort represent a very important opportunity for a targeted intervention.

There were 14 (14.29%) non-Hispanic Caucasian patients using statins and 84 (85.71%) not using statins. Thirty (22.90%) AA patients were on statins and 101 (77.10%) were not using statins (*p* = 0.1). AA patients had higher statin use in our study, but this difference did not reach statistical significance. However, we cannot exclude the possibility that higher statin usage by AA compared to non-Hispanic Caucasians could potentially contribute to the presence of higher comorbidities in AA. Several observational studies have found that statin use is associated with a lower risk of hepatocellular carcinoma (HCC) [1,34,35]. In one meta-analysis of ten studies with 1.6 million patients comparing patients' statin use, those who took statins had a 37% lower chance of developing HCC (odds ratio 0.63; 95% CI 0.52–0.76) [36]. This effect was most profound in East-Asian males with chronic hepatitis B, who are at high risk for the development of HCC. The effect has been far less dramatic in studies from the U.S. and Europe, where the proportion of patients with HCC due to chronic HBV infection is much lower. However, it is unclear whether this observed association is because of confounding effects of these influences [37].

Gallstones and cholecystectomy were found to increase the risk of primary liver cancer in a meta-analysis of 15 studies with over 4 million subjects [38]. The meta-analysis found that the odds ratio for developing liver cancer was 2.5 (95% CI 1.7–3.8) among patients with gallstones and was 1.6 (95% CI 1.3–2.0) among patients who had undergone a cholecystectomy. In our study we found higher incidence of gallstones in AA (42.75% vs. 29.6%, $p = 0.03$) but lower rates of cholecystectomy (10.23% vs. 19.14%, $p = 0.02$).

When we consider the tumor size, the AA group had significantly larger tumors at presentation compared to the non-Hispanic Caucasian group ($p < 0.02$). For AA, 40.46% of patients had HCC > 5 cm, whereas this was the case for 23.47% only of non-Hispanic Caucasians. These results are thus also consistent with the disparity seen in the inclusion in Milan criteria ($p < 0.01$). These results are consistent with the study presented by Ha et al., where the AA group was more likely to have advanced tumors at presentation (20.9% vs. 18.7, $p < 0.001$) and less likely to have tumors meeting Milan criteria (29.2% vs. 31.0%, $p < 0.01$) [39].

The reasons for these differences have been speculated on but are not really known. There are known differences in the access to healthcare between the two groups. In addition, social and economic factors have been known to play a significant role in the healthcare delivery between the two groups. One indicator for favorable outcomes in cancers is marriage [40–43]. This is a significant SES marker and can directly affect patient's ability to take care of himself. In our study, the marriage rates were comparable between the two groups. (26.53% vs. 25.9%).

Our work has several caveats for interpretation. Firstly, since our study is retrospective in nature, it holds the known biases associated with this type of study. Our future studies may include a prospective analysis of HCC incidence. Secondly, as mentioned, the level of clinical data available to us does not capture significant details that may affect the use of surgical therapy or survival. This includes medical comorbidities—the presence of chronic liver disease, etiologies such as HBV infection and aflatoxin food contamination, and the degree of liver cirrhosis; and information on the details of all treatments received. We also acknowledge that future studies should increase vigilance in obese patients, as it may pick up HCC otherwise missed by US.

Thirdly, the socioeconomic data we were able to collect did not fully capture the economic, educational, and social factors for individual patients. Lack of social support, density of specialists within a region, hospital volume, distance to care, and other unmeasured confounders may have influenced access to therapies. Lastly, we acknowledge that a larger sample size would increase the precision of our data, and future studies should include increasing sample size by doing a multicenter study. Nevertheless, to our knowledge, our analysis has discovered several new indices which are significantly different between the two ethnic groups of HCC patients in the US.

In conclusion, we have discovered several new and significant ethnical disparities in the presentation of HCC patients in the US (shown in Figure 1, below). Compared to non-Hispanic Caucasians, AA patients at presentation were older, were more likely to have modifiable metabolic risk factor such as diabetes, had larger HCCs, had higher AFP values, were more likely to have gallstones, and were more likely to not meet Milan criteria.

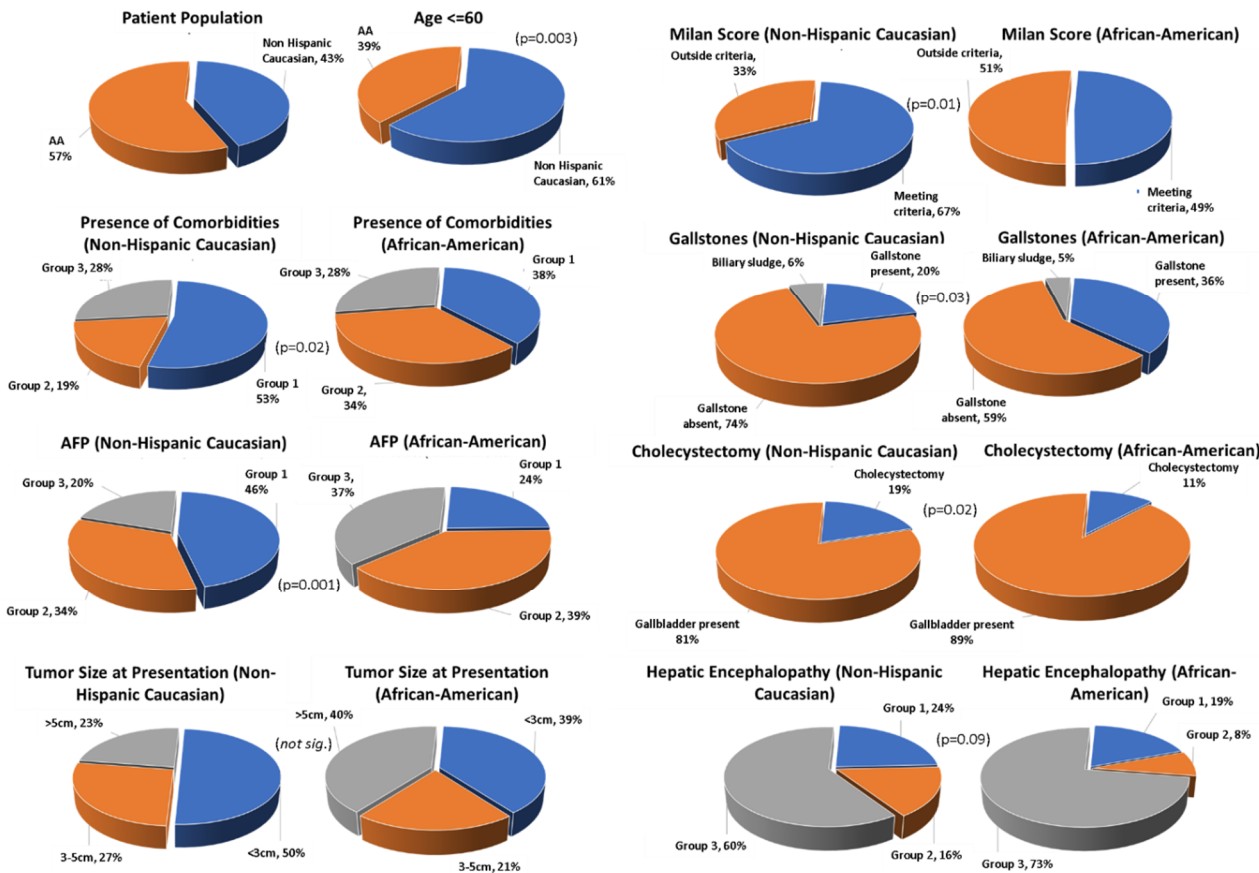

**Figure 1.** Differences between Caucasian and African American (AA) populations and disease presentation in HCC. Graphical data are summarized from results Section 3. Shown are *p*-values or non-significance (not. sig.). Left panel: African American vs. Caucasian group representation, age, presence of co-morbidities, AFP values, and tumor size at presentation. Right panel: Milan score, presence of gallstones, cholecystectomy, and hepatic encephalopathy.

**Author Contributions:** Conceptualization, K.K. and H.S.; methodology, K.K. and H.S.; software, K.K., P.J., R.S., K.P. and J.M.; validation, R.S., K.P. and J.M.; formal analysis, K.K., P.J., R.S., K.P. and J.M.; investigation, J.S.A., K.K., H.S. and P.J.; data curation, K.K., P.J. and H.S.; writing—original draft preparation, K.K. and H.S.; writing—review and editing, J.S.A., K.K., P.J., H.S., A.S. and R.S.; visualization, K.K., R.S., K.P., A.S. and J.M.; supervision, K.K. and H.S.; project administration, K.K. and H.S. All authors have read and agreed to the published version of the manuscript.

**Funding:** This research received no external funding.

**Institutional Review Board Statement:** The study was conducted according to the guidelines of the Declaration of Helsinki and approved by the Institutional Review Board of LSUHSC-S (IRB #: Study00001631, approved 4/2021).

**Informed Consent Statement:** Informed consent was obtained from all subjects involved in the study.

**Data Availability Statement:** Data are upon request from the author (HS).

**Acknowledgments:** We thank the LSUHSC Department of Medicine for supporting this project.

**Conflicts of Interest:** The authors declare no conflict of interest.

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
