# Peer review of "Clinical Presentation of Hepatocellular Carcinoma in African Americans vs. Caucasians: A Retrospective Analysis"

_pathophysiology, doi:10.3390/pathophysiology28030026_

Round 1

Reviewer 1 Report

Authors described that several new and significant ethnical disparities in the presentation of HCC patients in the US local area. Compared to non-Hispanic Caucasians, AA patients at presentation were older, had increased incidence of modifiable metabolic risk factor like diabetes, larger HCC size, higher AFP values, elevated incidence of gallstones and more likely to be outside Milan criteria.

  • The original concept of compare A A with Caucasian is unclear.

In the first place, the sample size is too small, and I think that the results of this study are very limited.

  • Since it does not show the ratio of the population of this area covered by the Louisiana state university to the population of AA and Caucasian, it is possible that a incidence of HCC has already been biased.
  • All evaluated parameters should be listed in Tables 1 and 2, such as number of HCV patients, etc. Also, there is too little numerical information in tables, and I wonder it is hiding on purpose.

The parameters used for multivariate and univariate cannot be evaluated because there were very few numerical values information on table 3 and 4.

Why is there no mention of HBV? I think that is very important parameter for incidence of HCC.

  • There is no reason for the selected parameters and are no reference articles related to the parameters.
  • Statistical method for multivariate analysis was not written.

Author Response

Reviewer 1

 We appreciate the reviewers comments and have responded to the critiques now including graphical data as requested.

1) The original concept of compare AA with Caucasian is unclear.

Response.   We agree and have now added the following lines to our introduction:

“We considered African Americans in comparison with Caucasians in our clinical practice. Nationally African Americans make up ~14% of the US population but are often under-represented in studies on race based presentation of disease and survival. However because our region has an approximate population makeup of ~50% AA and 50% white, we are better able to consider such contributions. Here we have evaluated such contributions to HCC. This study was conducted to evaluate the factors responsible for theses difference which can help us to serve our community better.”

2) Since it does not show the ratio of the population of this area covered by the Louisiana state university to the population of AA and Caucasian, it is possible that a incidence of HCC has already been biased.

Response.  We regret not having addressed this. Louisiana state university serves a population with predominantly 56% AA patients which is more similar to 1:1 than the national average of 14% African American. This is now stated in the introduction. Although the data are retrospective this study does at least reflect our trends which may be representative of national trends. Our future studies may include a prospective analysis of HCC incidence. Here, we have attempted here to compare factors responsible for the apparent different presentation of AA and Caucasian patients within our clinical practice. 

3) All evaluated parameters should be listed in Tables 1 and 2, such as number of HCV patients, etc. Also, there is too little numerical information in tables, and I wonder it is hiding on purpose.

Response.  We have now adjusted the tables again as requested. 

We recognize that hepatitis B data and aflatoxin food contaminant might be valuable but are not available and recognize this as a limitation of the paper. We have now stated this in the discussion section.

4) Used cut-offs to divide cohorts accordingly. Please explain why e.g. AFP 20ng/mL is such cut-off… Please provide rational of AFP grouping into 1,2,3. please cite references. Do this, provide this for all included parameters, if applicable.

Response.  We use 20 ng/ml as cut off for AFP per our institutional lab ‘normal’ range. We also selected  AFP > 200 as a range because it is is associated with poor outcome. Grouping of AFP values allows for easier visualization and comparison of data. Groups were determined according to the understanding that AFP values between 0-20 in adults is considered normal while AFP values >200 are considered diagnostic for HCC.

We now include the following statements:

“Bialecki states that: “”AFP >400–500 ng/ml is considered diagnostic for HCC, although fewer than half of patients may generate levels that high’. Zhang et al., in a meta-analysis showed that AFP in excess of 200 ng/ml was associated with HCC. Therefore, we also selected values of AFP >200 as an upper cut-off value’

We have now added the following citations in support of this.

Zhang J, Chen G, Zhang P, Zhang J, Li X, Gan D, Cao X, Han M, Du H, Ye Y. The threshold of alpha-fetoprotein (AFP) for the diagnosis of hepatocellular carcinoma: A systematic review and meta-analysis. PLoS One. 2020 Feb 13;15(2):e0228857. doi: 10.1371/journal.pone.0228857. PMID: 32053643; PMCID: PMC7018038.

Bialecki, Eldad S, and Adrian M Di Bisceglie. “Diagnosis of hepatocellular carcinoma.” HPB : the official journal of the International Hepato Pancreato Biliary Association vol. 7,1 (2005): 26-34. doi:10.1080/13651820410024049

Reviewer 2 Report

I read with interest the work of Semat and Colleagues concerning the different clinical presentation of hepatocellular carcinoma in African American vs Caucasian.

229 patients (between 2008 and 2018) are not a large number of patients to draw definitive conclusions; it would be useful to increase the sample size by involving more centers.

Despite this, there is a lack of important data to characterize the population with HCC, such as alcohol consumption, weight, how many patients were HCV, HBsAg or HBcAb positive and antiviral therapies that, as is known, impact on the natural history of the disease. These parameters have a strong impact on the results.

Regarding the comorbidities, I do not understand how hypertension is not associated with group 2 which includes dyslipidemia and diabetes. Since the authors mention metabolic SD in the discussion it would be important to describe it in the target population.

The Authors described  a higher number of statin users in AA, perhaps because they had a major presence of  comorbidities? Under discussion is not mentioned. 

The authors define the stage of the tumor by BCLC but do not show in the table the presence of cirrhosis vs non-cirrhosis and the degree of cirrhosis while defining the degree of encephalopathy well. 

The US shows the presence of HCC in less than 50% of patients and many patients had tumor sizes> 3 cm. Were all patients under surveillance? only those with cirrhosis? was the diagnosis difficult because the patients were obese?

Author Response

Reviewer 2

We appreciate the reviewers comments and have now revised the study to include the responses below as well as visual descriptions of the data as requested.

1) 229 patients (between 2008 and 2018) are not a large number of patients to draw definitive conclusions; it would be useful to increase the sample size by involving more centers

Response.  We agree that increasing the sample size by doing a multicenter study will be the next step. We agree that increasing the sample size would be useful as it would increase the precision of our data.  We have now mentioned this in the discussion as a potential drawback of the study.

2) Despite this, there is a lack of important data to characterize the population with HCC, such as alcohol consumption, weight, how many patients were HCV, HBsAg or HBcAb positive and antiviral therapies that, as is known, impact on the natural history of the disease. These parameters have a strong impact on the results.

Regarding the comorbidities, I do not understand how hypertension is not associated with group 2 which includes dyslipidemia and diabetes. Since the authors mention metabolic SD in the discussion it would be important to describe it in the target population.

Response.  Please review the tables. Alcohol consumption data was included in table 3 in connection with HCV infection. We agree that general social history of alcohol consumption, weight, and antiviral therapy use could be included in future studies. Hypertension is included in group 3 of comorbidities.  

3)  The authors define the stage of the tumor by BCLC but do not show in the table the presence of vs non-cirrhosis and the degree of cirrhosis while defining the degree of encephalopathy well.

Response.  All the included patients were cirrhotic. We did not have data on degree of cirrhosis 

Round 2

Reviewer 1 Report

It is well written and response to our revise requests.

Reviewer 2 Report

The work can be published after the changes I have reviewed